# Intravenous versus Oral Step-Down for the Treatment of *Staphylococcus aureus* Bacteremia in a Pediatric Population

**DOI:** 10.3390/pharmacy10010016

**Published:** 2022-01-15

**Authors:** Sarah Grace Gunter, Mary Joyce B. Wingler, David A. Cretella, Jamie L. Wagner, Katie E. Barber, Kayla R. Stover

**Affiliations:** 1Department of Pharmacy, Grandview Medical Center, Birmingham, AL 35243, USA; sarah.gunter@grandviewhealth.com; 2Department of Antimicrobial Stewardship, University of Mississippi Medical Center, Jackson, MS 39216, USA; dcretella@umc.edu; 3Department of Pharmacy Practice, University of Mississippi School of Pharmacy, Jackson, MS 39216, USA; jwagner@umc.edu (J.L.W.); kbarber@umc.edu (K.E.B.); kstover@umc.edu (K.R.S.)

**Keywords:** *Staphylococcus aureus*, bloodstream infection, pediatric, bacteremia

## Abstract

Limited data are available regarding optimal antimicrobial therapy for *Staphylococcus aureus* bacteremia (SAB) in pediatric patients. The purpose of this study was to assess clinical characteristics and outcomes associated with intravenous (IV) versus oral step-down treatment of pediatric SAB. This study evaluated patients aged 3 months to 18 years that received at least 72 h of inpatient treatment for SAB. The primary endpoint was 30-day readmission. Secondary endpoints included hospital length of stay and inpatient mortality. One hundred and one patients were included in this study. The median age was 7.9 years. Patients who underwent oral step-down were less likely to be immunocompromised and more likely to have community-acquired SAB from osteomyelitis or skin and soft tissue infection (SSTI). More patients in the IV therapy group had a 30-day readmission (10 (25.6%) vs. 3 (5.3%), *p* = 0.006). Mortality was low (5 (5%)) and not statistically different between groups. Length of stay was greater in patients receiving IV therapy only (11 vs. 7 days, *p* = 0.001). In this study, over half of the patients received oral step-down therapy and 30-day readmission was low for this group. Oral therapy appears to be safe and effective for patients with SAB from osteomyelitis or SSTIs.

## 1. Introduction

Known for its virulence, *Staphylococcus aureus* can cause significant morbidity and mortality in both pediatric and adult patients. This is particularly true for those with invasive disease, such as bloodstream or endovascular infections [1]. Mortality appears to be lower in the pediatric population than in adults [1,2,3]; however, concern for morbidity, including disease and treatment-related complications, remains [2,4,5].

Extensive research has been conducted on the treatment of adult patients with *S. aureus* infections [1,6,7,8,9,10,11,12]. In contrast to the abundant literature on adult patients with *S. aureus* bacteremia (SAB), there is a paucity of evidence to guide practitioners in the treatment of pediatric patients. In the majority of pediatric patients, SAB is associated with bone and joint infections [3,4], whereas adult infections are often related to catheters or other medical devices [1]. These differences in etiology between adults and children make it difficult to know if adult treatment recommendations should be applied in children. 

The use of oral antibiotics for SAB has yet to be validated by prospective, randomized controlled trials and has only been evaluated by a select few studies, most of which involved subgroup analyses of larger studies where the predominant number of patients were nonbacteremic [5,13,14,15]. Two retrospective [16,17] and one prospective cohort [18] studies that evaluated oral therapy for adult patients with SAB demonstrated low incidence of complications and similar mortality to patients who received a full course of intravenous therapy. It is important to note that these studies included a majority of patients with low-risk methicillin-susceptible SAB, which limits applicability in settings of higher acuity, more serious infections, or where methicillin resistance may be more prevalent. Only one study has evaluated oral therapy for SAB in children [5]. This study evaluated patients with bacteremic osteoarticular infections and revealed that short-term IV treatment followed by a course of oral therapy may be reasonable in certain pediatric patients.

Possibly as a result of the limited data available, one challenge with SAB in children is the lack of consistency in treatment across institutions. A survey of pediatric infectious diseases (ID) providers revealed wide variability in the treatment of SAB. For patients with osteomyelitis-associated SAB, 50% of providers switched to oral antibiotics for nonpersistent bacteremia and 22% switched after clinical improvement. For nonpersistent, non-osteomyelitis-associated SAB, 54% of providers switched patients to oral antibiotics after receiving empiric vancomycin [14].

The practices at our institution for selecting IV or oral therapy for SAB are heterogeneous in pediatric patients. The goal of this study was to compare the clinical characteristics and outcomes of pediatric patients receiving a full course of IV therapy to those receiving oral step-down treatment for SAB from any source.

## 2. Materials and Methods

This retrospective cohort was conducted at a tertiary academic medical center and evaluated patients admitted between 1 June 2012 and 19 November 2018. Patients with a positive blood culture were identified using TheraDoc^®^ Clinical Surveillance Software (Premier, Inc., Charlotte, NC, USA) and included if they were between the ages of 3 months and 18 years, had a blood culture positive for *S. aureus*, and received at least 72 h of inpatient IV treatment. Exclusion criteria were as follows: pregnancy, death within 72 h of initial culture, hospice or palliative care, and polymicrobial bacteremia. For patients with multiple incidences of bacteremia within the study period, only the first qualifying admission was included. Data was collected using Research Electronic Data Capture (REDCap™) (Vanderbilt University, Nashville, TN, USA) [19]. Positive cultures were identified initially using a BD BACTEC FX^®^ instrument (Becton Dickinson, Franklin Lakes, NJ, USA). Prior to 2016, samples demonstrating Gram-positive cocci on Gram stain were identified using biochemical testing and then confirmed using Vitek2 (bioMerieux, Durham, NC, USA). In 2016, this process was changed to include the identification of Gram-positive cocci via a Biofire FilmArray^®^ Blood Culture ID Panel (Salt Lake City, UT, USA), then confirmed using matrix-assisted laser desorption–ionization time-of-flight mass spectrometry with VitekMS and Vitek2 (bioMerieux, Durham, NC, USA).

### 2.1. Endpoints

The primary endpoint was the rate of 30-day readmissions for patients receiving a full course of IV treatment compared to those who received oral step-down therapy. Secondary endpoints included time to microbiological cure, infection-related length of stay, total hospital length of stay, total duration of therapy, clinical failure at 90 days, 90-day readmission rates, rates of pediatric intensive care unit (PICU) admission, attributable inpatient mortality, all-cause inpatient mortality, and rates of adverse drug reactions (ADRs).

### 2.2. Definitions

Oral step-down treatment was defined as a transition from IV antistaphylococcal therapy to oral antistaphylococcal therapy after a minimum of 72 h of IV treatment. Community-acquired infection was defined as initial positive blood cultures that were drawn within 48 h of admission, whereas hospital-acquired infection was defined as positive cultures drawn ≥48 h after admission. Healthcare-associated infection was defined as a community-onset infection plus the presence of a medical device in situ.

Microbiological cure was defined as negative blood cultures following the initial positive culture; any further blood cultures within one week of the first negative culture were required to remain negative to confirm microbiological cure (i.e., microbiological cure was not demonstrated if clearance of one blood culture was documented, but a following culture the next day returned positive).

Clinical failure at 90 days was defined as a composite of readmission within 90 days and/or recurrence of bacteremia. Recurrence of bacteremia was defined as a new culture positive for *S. aureus* that was separated by at least 7 days (but no more than 30 days) from the last positive blood culture for *S. aureus* with at least one negative blood culture in the interim period. Reinfection was defined as a new culture positive for *S. aureus* that was separated by at least 30 days from the last positive blood culture for *S. aureus* with at least one negative blood culture in the interim period.

Persistent bacteremia was defined as continuously positive blood cultures for >3 days from the initial positive culture. Postinfection length of stay was defined as the number of inpatient days beginning with the date of the first positive blood culture. Attributable inpatient mortality was defined by a blood culture positive for *S. aureus* at the time of death, cause of death listed as *S. aureus* in the medical record, or initial blood culture positive for *S. aureus* within 14 days of death and no other attributable cause. Readmission attributable to SAB was defined as reason for admission documented as SAB (recurrence or reinfection) or, additionally, complications related to SAB treatment (i.e., PICC line complications for patients receiving outpatient IV treatment, ADRs attributed to antistaphylococcal therapy, need for further surgical debridement, etc.).

Complicated infection was defined as those involving deep tissue abscesses, pulmonary infections, infective endocarditis, no defined focus of infection, infection involving sepsis (hemodynamic instability requiring fluid bolus or inotropes), PICU admission, and/or multiple (>1) noncontiguous sites of infection. In contrast, simple infections were those that only involved a single site of infection (or contiguous sites, if multiple) and that did not meet the previously described criteria for complex infection.

Source control was defined as surgical intervention, debridement, and removal of infected hardware or central line during hospitalization. Device-related infections were defined as those from indwelling central venous catheters and/or orthopedic hardware.

Immunocompromising conditions were defined as follows: immunosuppressive pharmacotherapy (including biologic medications such as rituximab, abatacept, adalimumab, infliximab, cancer chemotherapy within the past 6 months, or >14 days of systemic corticosteroid use at doses greater than 1 mg/kg prednisone equivalent daily, etc.), human immunodeficiency virus (HIV) infection, neutropenia (defined as an absolute neutrophil count <500 cells/mm^3^), other immunodeficiency syndromes (including severe combined immunodeficiency, common variable immunodeficiency, and Winskott–Aldrich syndrome) [2,20,21], solid organ transplantation within 1 year, and hematopoietic stem cell transplantation (HSCT) within 1 year.

### 2.3. Statistical analysis

Categorical data were assessed using a chi-square test or Fisher’s exact test, as appropriate. Continuous data were analyzed using a Student’s t-test for parametric data or Mann–Whitney U test for nonparametric data. A *p*-value ≤ 0.05 was considered statistically significant. Statistical analyses were conducted using Microsoft Excel^®^ and IBM SPSS Statistics 24.0 (IBM Corp, Armonk, NY, USA).

## 3. Results

Two hundred thirty-seven patients were screened for enrollment, and 101 patients were included in the analysis (42 IV therapy; 59 oral step-down) (Figure 1). Patients who received IV therapy exclusively were more likely to be admitted to an ICU, have an immunocompromising condition, a healthcare-associated or hospital-acquired infection, or endocarditis and device-related infection (Table 1 and Table 2). Conversely, patients that underwent oral step-down were more likely to have community-acquired infections and a primary diagnosis of osteomyelitis or skin and soft tissue infection (SSTI) (Table 1). Methicillin resistance was similar between groups (60.5% IV group vs. 51.7% oral step-down; *p* = 0.313).

Information about antimicrobial therapy can be found in Table 3. During the course of therapy, most patients received more than one antistaphylococcal agent, with a median of two agents (IQR, 1–3) in the IV group and three agents (IQR, 2–4) in the oral step-down group (*p* = 0.006). Significantly more patients in the IV group received concomitant (nonstaphylococcal) antibiotic therapy (61.9% vs. 32.8%; *p* = 0.004). Patients with osteomyelitis (n = 33), septic arthritis (n = 7), and infective endocarditis (n = 5) had median treatment durations of 42.0 days. Those with device-associated infections (n = 23) were treated with a median of 15.0 total days of therapy, and bacteremic SSTIs (n = 8) were treated for a median of 13.5 days. This led to a total duration of therapy that was longer in the oral group compared to the IV group (median 33.0 days vs. 16.0 days; *p* = 0.001).

Thirty-day readmission occurred in 13 (13.5) patients: 10 (25.6%) in the full-course IV therapy group and 3 (5.3%) in the oral step-down group (Table 3). Of the 10 patients in the IV group who were readmitted, 40% (n = 4) had a diagnosis of endocarditis. Source control was performed in most patients in both groups (8/10 in the IV group and 3/3 in the oral step-down group). The majority of patients who were readmitted had an ID consult during the initial admission (11/13, 84.6%) and were considered complicated SAB (11/13, 84.6%). Of the readmitted patients, the median duration of bacteremia was 6 days. No readmitted patients had recurrent bacteremia; however, two patients experienced reinfection, and both were in the IV group.

The median length of stay was 11.0 days (IQR, 8.0–21.0) in the IV group and 7.0 days (IQR, 5.0–11.0) in the oral step-down group (*p* = 0.001). All-cause inpatient mortality occurred in four patients (9.5%) in the IV group compared to one (1.7%) in the oral step-down group (*p* = 0.160). A further breakdown of secondary endpoints can be seen in Table 3. Treatment and outcomes stratified by methicillin resistance can be found in Table 4.

## 4. Discussion

In a time when oral step-down therapy is growing increasingly popular, questions remain regarding the utility of oral therapy in SAB, especially in children. The management of SAB at our institution is diverse with no clear and agreed-upon standard or guidance. Our pediatric patients with SAB experienced a high rate of complicated infection (66.3%) and clinical failure (54.5%), driven by persistently positive blood cultures. Despite this, over half of patients were converted to oral therapy.

In our study, the broad majority of patients had SAB secondary to bone, joint, or muscle infections (84.5%). This is expected based on etiology studies in pediatrics [3,4] and is similar to the study by Kouijzer et al. [17]. Other studies from Bupha-Intr and Willekens and colleagues evaluated oral therapy for adults with SAB and revealed that the majority of the patients had line-associated infections [16,18].

Patients who received oral step-down therapy in our study experienced a low rate of 30-day readmission (5.3%) and no SAB-attributable mortality. This is similar to studies by Kouijzer (no relapse, 6.6% mortality), Bupha-Intr (1% relapse in 90 days, no attributable mortality), and Willekens (2.2% 90-day relapse, 2.2% mortality) [16,17,18]. In addition, these results mirror those seen in previous studies of bacteremic *S. aureus* osteomyelitis [5,14,22].

Length of hospital stay in our study was shorter in the group of patients treated with oral step-down therapy (7 vs. 11 days), but total treatment duration was longer (33 vs. 16 days). Our findings mirror those of McNeil et al. in their review of bacteremic osteomyelitis patients. Their analysis found that the length of hospital stay was shorter in the oral step-down group, with a length of stay nearly matching our cohort (median of 8.5 days in the oral step-down group vs. 11 days in the IV group; *p* = 0.006) [5]. Similar to our study, Willekens and Kouijzer and colleagues found longer lengths of stay in the IV groups as compared to the oral step-down groups (median 19 vs. 8 days (*p* < 0.01) and median 26 vs. 17 days (*p* = 0.001), respectively) [17,18]. In contrast to our study, there was no difference in the total duration of treatment between groups in the studies by Kouijzer (median 45 days for both), Willekens (median 15 days for both), and Bupha-Intr (median 16 vs. 14 days) [16,17,18]. These differences in results likely reflect differing underlying sources of infection between pediatrics and adults but could also reflect the difficulty of outpatient IV antimicrobial therapy, particularly in pediatric patients.

Methicillin resistance was common in our study, observed in more than half of the patients in each group. This is a notably larger percentage than has been reported in previous studies of SAB in pediatric patients (0–44%) [3,5,14,22,23,24] or in other studies of oral therapy in adults with SAB (3–18%) [16,17,18]. Although there is conflicting evidence regarding the clinical significance of methicillin resistance on treatment outcomes, some data suggest a higher probability of clinical failure and mortality in patients with MRSA compared to MSSA [24,25]. Overall mortality in the current study was similar to that reported previously, and no organism-specific effects on outcomes were observed (Table 4) [24,25].

In this cohort, 32% of these patients received an additional anti-MRSA antibiotic; 21 patients received clindamycin instead of vancomycin within the initial 48 h. This appeared to reflect a practice of beginning initial therapy with clindamycin and switching to vancomycin when bacteremia was identified. Unfortunately, it is difficult to compare antibiotic selection among studies, as not all previous studies have detailed the agents used in treatment [5,13]. Munro et al. reported that their patients received primarily antistaphylococcal penicillins (flucloxacillin; 93%), followed by vancomycin (63%) [3]. Notably, only 10.2% of isolates were MRSA. Hamdy et al., who evaluated the epidemiology and risk factors associated with the treatment of MRSA bacteremia, reported that 88.4% (205/232) received vancomycin within the first 48 h [2].

The primary limitations of this study stem from its retrospective, unmatched design. Patients receiving exclusively IV therapy and those receiving oral step-down differed significantly in observed characteristics but were also at risk for confounding for other markers of disease severity. The current study did not take into account emergency department (ED) visits that did not result in an admission. Although an admission likely signifies a more serious complication with therapy, it is possible that accounting for ED visits would provide a more comprehensive view of treatment success or failure. It has been suggested that pediatric patients discharged with outpatient IV antibiotics are at a higher risk of complications associated with their treatment course [22]. Additionally, our sample was small and heterogeneous. Lastly, this study evaluated patients during a 7-year period, and differences in SAB management may have changed over time. The effect of this was not assessed and may have impacted clinical outcomes.

Oral step-down therapy has many potential advantages including reduced healthcare resource utilization and reduced risk of IV therapy complications, which makes these results potentially significant. For this reason, many clinicians and investigators are exploring oral step-down therapy for serious systemic infections. Emerging data for endocarditis and bone and joint infections derived from adults suggest that oral therapy can be effective, but *S. aureus*, particularly MRSA, is poorly represented in those studies [26,27]. In a diverse cohort of pediatric patients with SAB, we observed a significant portion of patients that received oral step-down antimicrobial therapy and these patients had a low rate of readmission and shorter hospital length of stay compared to those receiving only IV therapy. Despite a high rate of MRSA, we did not observe a difference in outcomes based upon susceptibilities or antimicrobial used. Our data suggest that oral step-down therapy may be a reasonable treatment option in some pediatric SAB patients, notably those with bone or SSTI-derived bacteremia. This strategy may not be suitable for those with hospital-acquired infection or endocarditis, but further study may be needed to identify individual risk factors for treatment failure.

## 5. Conclusions

In summary, we conducted a retrospective assessment of pediatric SAB management and outcomes. Patients who underwent oral step-down therapy had low rates of readmission or recurrence and a decreased length of stay as compared to patients who received a full course of IV therapy. Oral step-down therapy appears to be a reasonable alternative to prolonged IV treatment for pediatric SAB, but further literature may help identify ideal patients and risk factors for treatment failure.

## Figures and Tables

**Figure 1 pharmacy-10-00016-f001:**
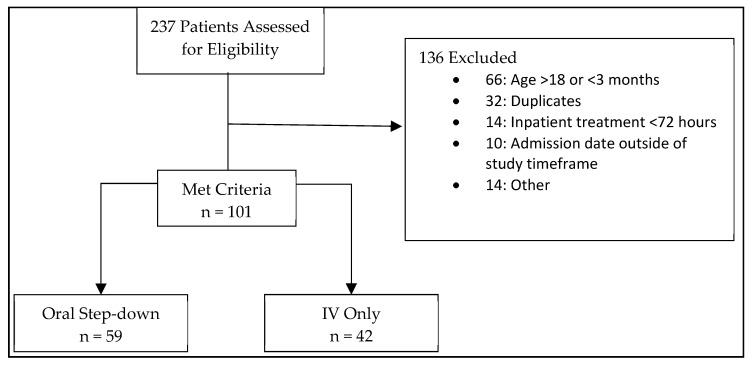
Patient Inclusion/Exclusion.

**Table 1 pharmacy-10-00016-t001:** Baseline characteristics.

Characteristic, n (%) or Median (IQR)	All	IV Only	Oral Step-Down	*p*-Value
	(n = 101)	(n = 42)	(n = 59)	
Age, years	7.9 (3.0, 12.2)	9.6 (5.6, 13.0)	5.7 (2.3, 11.9)	0.071
Male	53 (52.5)	21 (50.0)	32 (54.2)	0.691
Race				0.474
Black	48 (47.5)	17 (40.5)	31 (52.5)	
White	46 (45.5)	22 (52.4)	24 (40.7)	
Other	7 (6.9)	3 (7.1)	4 (6.8)	
Comorbidity				
None	57 (56.4)	24 (55.8)	33 (56.9)	0.813
Premature Birth	16 (15.8)	5 (11.9)	11 (18.6)	0.418
Chronic Lung Disease	14 (13.9)	4 (9.5)	10 (16.9)	0.385
Eczema	13 (12.9)	5 (11.9)	8 (13.6)	1.000
Heart Disease	10 (9.9)	6 (14.3)	4 (6.8)	0.312
Neuromuscular Disease	4 (4.0)	1 (2.4)	3 (5.1)	0.639
End-Stage Renal Disease	3 (3.0)	3 (7.0)	0 (0)	0.069
Diabetes Mellitus	1 (1.0)	0 (0)	1 (1.7)	1.000
Immunosuppression ^	87 (86.1)			
None	8 (7.9)	29 (69.0)	58 (98.3)	<0.001
Immunosuppressive Pharmacotherapy *		8 (19.0)	0 (0)	0.001
Neutropenia	8 (7.9)			
Hematopoietic Stem Cell Transplant **	3 (3.0)	7 (16.7)	1 (1.7)	0.010
		3 (7.1)	0 (0)	0.074
Onset of Infection				< 0.001
Community	63 (62.3)	13 (31.0)	50 (84.7)	
Hospital	13 (12.9)	11 (26.2)	2 (3.4)	
Healthcare-Associated	25 (24.7)	18 (42.9)	7 (11.9)	
Duration of Fever, Days	3.0 (2.0, 5.5)	3.0 (1.0, 6.0)	3.0 (2.0, 5.3)	0.688
WBC Count, Initial (n = 100)	12.7 (8.8, 18.7)	10.5 (5.1, 17.1)	13.9 (11.2, 21.6)	0.011
CRP				
Initial (n = 69)	15.4 (5.2, 27.0)	8.0 (3.7, 27.5)	16.6 (6.2, 27.0)	0.291
Max (n = 71)	19.0 (7.0, 32.8)	18.6 (4.1, 30.5)	19 (7.5, 33.4)	0.332
ESR, Initial (n = 60)	57.0 (36.3, 87.8)	51.0 (25.0, 91.0)	59.5 (37.0, 87.3)	0.374
Time to Therapy (hours)	4.0 (1.0, 14.0)	3.5 (1.0, 13.0)	5.0 (1.0, 15.0)	0.741
Primary Focus of Infection				
Osteomyelitis	33 (32.6)	5 (11.9)	28 (47.4)	< 0.001
Device	23 (22.8)	18 (42.9)	5 (8.5)	< 0.001
SSTI	8 (7.9)	0 (0)	8 (13.6)	0.020
Pneumonia	7 (6.9)	5 (11.9)	2 (3.4)	0.132
Septic Arthritis	7 (6.9)	3 (7.1)	4 (6.8)	1.000
Deep Tissue Abscess	6 (5.9)	1 (2.4)	5 (8.5)	0.236
Endocarditis	5 (4.9)	5 (11.9)	0 (0)	0.005
Pyomyositis	6 (5.9)	1 (2.3)	5 (8.6)	0.236
Unknown	6 (5.9)	4 (9.5)	2 (3.4)	0.397
Multiple Foci of Infection	47 (46.5)	15 (35.7)	32 (54.2)	0.073
Methicillin Resistance	56 (55.4)	26 (60.5)	30 (51.7)	0.313

IQR = Interquartile range; IV = Intravenous; WBC = White blood cell; CRP = C-reactive protein; ESR = Erythrocyte sedimentation rate; SSTI = Skin/soft tissue infection. * Within 6 months; ** within 1 year; ^ patients may have more than one immunosuppressing condition.

**Table 2 pharmacy-10-00016-t002:** Outcomes.

Endpoint, n (%) or	All	IV Only	Oral Step-Down	*p*-Value
Median (IQR)	(n = 101)	(n = 42)	(n = 59)	
30-Day Readmission *	13/96 (13.5)	10 (25.6)	3 (5.3)	0.006
90-Day Readmission *	18/96 (18.6)	13/39 (33.3)	5/57 (8.8)	0.003
Inpatient Mortality				
All-Cause	5 (5.0)	4 (9.5)	1 (1.7)	0.160
Attributable	1 (1.0)	1 (2.3)	0 (0)	0.426
Reinfection	4/97 (4.1)	4/40 (10.0)	0/57 (0)	0.026
Recurrence	3/98 (3.1)	2/40 (5.0)	1/58 (1.7)	0.570
Persistent Bacteremia	49/100 (49.0)	24/42 (57.1)	25/58 (43.9)	0.224
Clinical Failure—Composite	55 (54.5)	28 (65.1)	27 (46.6)	0.064
Length of Stay				
Total	9.0 (6.0, 18.0)	11.0 (8.0, 21.0)	7.0 (5.0, 11.0)	0.001
Postinfection	8.0 (6.0, 15.0)	10.0 (7.0, 16.0)	7.0 (5.0, 11.0)	0.003
Duration of Bacteremia	4.0 (3.0, 5.0)	4.0 (3.0, 6.0)	4.0 (3.0, 5.0)	0.547
PICU Admission	37 (36.6)	21 (50.0)	15 (25.4)	0.009
Complicated Bacteremia	67 (66.3)	29 (69.0)	38 (64.4)	0.674
Infectious Diseases Consultation	82 (81.2)	31 (73.8)	51 (86.4)	0.127

PICU = Pediatric intensive care unit. * Among survivors.

**Table 3 pharmacy-10-00016-t003:** Antistaphylococcal therapy.

Characteristic, n (%) or Median (IQR)	All	IV Only	Oral Step-Down	*p*-Value
	(n = 101)	(n = 42)	(n = 59)	
**Primary IV Therapy**				
Vancomycin	50 (50.5)	26 (61.9)	24 (41.4)	0.044
Nafcillin/Oxacillin	21 (20.8)	5 (11.6)	16 (27.6)	0.051
Cefazolin	9 (8.9)	4 (9.3)	5 (8.5)	0.720
Clindamycin	17 (16.8)	5 (11.6)	12 (20.7)	0.229
Ceftriaxone	2 (2.0)	0 (0)	2 (3.4)	0.506
Other *	2 (2.0)	2 (4.7)	0 (0)	0.179
**Primary PO Therapy ^**				
Clindamycin	29 (28.7)	-	29 (49.2)	-
Cephalexin	21 (20.8)	-	21 (35.6)	-
Sulfamethoxazole- Trimethoprim	4 (4.0)	-	4 (6.7)	-
Linezolid	3 (3.0)	-	3 (5.1)	-
Other *	3 (3.0)	-	3 (5.1)	-
Doxycycline	1 (1.0)	-	1 (1.7)	-
**Miscellaneous**				
Number of Antistaphylococcal Agents	2 (2, 4)	2 (1, 3)	3 (2, 4)	0.006
Concomitant Antibiotic Therapy	45 (44.6)	26 (61.9)	19 (32.8)	0.004
Duration of Therapy, in Days				
Total	30.0 (14, 43.0)	16.0 (14.0, 42.0)	33.0 (25.5, 48.0)	0.001
IV	12.0 (5.5, 24.5)	16.0 (14.0, 42.0)	6.0 (4.0, 12.3)	<0.001
PO	25.0 (14.0, 42.0)	-	26.0 (14.0, 42.0)	-

IV = Intravenous; PO = Oral; * Other: IV–cefepime (n = 1); meropenem (n = 1); PO–amoxicillin-clavulanate (n = 1); amoxicillin (n = 1), cefdinir (n = 1). ^ Note: 2 patients each received 2 PO agents.

**Table 4 pharmacy-10-00016-t004:** Treatment and outcomes stratified by methicillin resistance.

Endpoint, n (%) or Median (IQR)	MSSA	MRSA	*p*-Value
	(n = 45)	(n = 56)	
Onset of Infection			
Community	29 (64)	34 (61)	0.837
Hospital	7 (16)	6 (11)	0.556
Healthcare-Associated	9 (20)	16 (28)	0.361
Primary Focus of Infection			
Osteomyelitis	13 (29)	20 (36)	0.526
Device	9 (20)	14 (25)	0.637
SSTI	4 (9)	4 (7)	1.000
Pneumonia	2 (4)	5 (9)	0.457
Septic Arthritis	4 (9)	3 (5)	0.697
Deep Tissue Abscess	4 (9)	2 (4)	0.403
Endocarditis	1 (2)	4 (7)	0.378
Pyomyositis	3 (7)	3 (5)	1.000
Unknown	5 (11)	1 (2)	0.086
Oral Step-down Performed	28 (62)	30 (54)	0.2674
Primary IV Therapy—MRSA			-
Vancomycin	-	43 (77)	
Clindamycin	-	11 (20)	
Other	-	2 (4)	
Primary IV Therapy—MSSA			-
Vancomycin	7 (16)	-	
Antistaphylococcal Penicillins	15 (33)	-	
Cefazolin	9 (20)	-	
Clindamycin	6 (13)	-	
Other	8 (18)	-	
Primary PO Therapy—MRSA ^			-
Clindamycin	-	23 (72)	
Sulfamethoxazole-Trimethoprim	-	4 (13)	
Doxycycline	-	1 (3)	
Linezolid	-	2 (6)	
Other	-	2 (6)	
Primary PO therapy—MSSA			-
Cephalexin	19 (66)	-	
Clindamycin	6 (21)	-	
Linezolid	1 (3)	-	
Other	3 (8)	-	
Duration of Therapy, in Days	20 (18–38)	42 (16–44)	0.018
Persistent Bacteremia	18 (40)	31 (55)	0.162
Duration of Bacteremia, in Days	3 (2–5)	4 (3–6)	0.062
30-Day Readmission *	6/43 (13)	7/53 (13)	1.000
90-Day Readmission *	10/43 (22)	8/53 (14)	0.436
Inpatient Mortality			
All-Cause	2 (4)	3 (5)	1.000
Attributable	0 (0)	1 (2)	1.000

* Among survivors. ^ Note: 2 patients each received 2 PO agents.

## Data Availability

The data presented in this study are available on request from the corresponding author.

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
