# Peer review of "Intravenous versus Oral Step-Down for the Treatment of Staphylococcus aureus Bacteremia in a Pediatric Population"

_pharmacy, 2022, doi:10.3390/pharmacy10010016_

Round 1

Reviewer 1 Report

In this work the authors have studied the treatment in Staphylococcus aureus bacteremia in a pediatric population from the view of intravenous to oral step-down. The study is very interesting because oral step down therapy seems to be  reasonable alternative to prolonged IV treatment with many advantages, but more data are needed to establish the conclusion. The study as authors mentioned has many limitations, retrospective design, small sample, population heterogeneity. Moreover, its an interesting study and the results have to be published as a reference for next similar studies. My suggestion is to be accepted after a better presentation of the literature including some additional references like:

Intravenous to Oral Switch in Complicated Staphylococcus aureus Bacteremia Without Endovascular Infection: A Retrospective Single-Center Cohort Study.

Ilse J E Kouijzer 1, Eline J van Leerdam et al., 2021.

Clin Infect Dis  2021 Sep 7;73(5):895-898. doi: 10.1093/cid/ciab156

Author Response

We have added this reference and a couple others that are relevant and revised the intro and discussion to better present the literature.

Reviewer 2 Report

The results are interesting. But the author needs to write the introduction and conclusion properly. Very insufficient writing in the introduction and conclusion. 

Author Response

We have made several revisions throughout the introduction and conclusion. If you had specific items within those sections you would like to see us revise, please let us know.  

Reviewer 3 Report

In this paper, the authors evaluated clinical characteristics and outcomes associated with intravenous versus oral step-down treatment of pediatric Staphylococcus aureus bacteremia. Overall, the results could be valuable for other researchers and physicians in this field. Some comments were suggested as follows.
1. Please avoid large clusters of references and disperse clusters larger than 5 references to a relevant place in the text and/or delete some of them.
“Mortality appears to be much lower in the pediatric population than in adults; however, concern for morbidity remains [1-6].”
“… there is a paucity of evidence to guide practitioners in the treatment of pediatric patients [1, 7-13].”
“… whereas adult infections are often related to catheters or other medical devices [1-5]”
2. Statistical analysis: “Continuous data were analyzed using a Student’s t-test or Mann-Whitney U test.” Please specify when the Mann-Whitney U test is used in this work.
3. The discussion section is just a mere summary of the results previously described. Instead, authors should put these results in context, compare their findings with the previous publications, discussing their significance.
4. Actually, this work involved a comparatively small number of pediatric patients (N = 101 in total). Therefore, these findings should be considered with extreme caution.

Author Response

Please find our responses below.

Please avoid large clusters of references and disperse clusters larger than 5 references to a relevant place in the text and/or delete some of them.
“Mortality appears to be much lower in the pediatric population than in adults; however, concern for morbidity remains [1-6].”

We have dispersed these references to eliminate the larger cluster of references.

“… there is a paucity of evidence to guide practitioners in the treatment of pediatric patients [1, 7-13].”

In this instance, we feel that the larger number of citations supports the referenced statement, so we have chosen to leave this grouping as is.

“… whereas adult infections are often related to catheters or other medical devices [1-5]”

Some references have been removed here and only the most relevant have been cited.

  1. Statistical analysis: “Continuous data were analyzed using a Student’s t-test or Mann-Whitney U test.” Please specify when the Mann-Whitney U test is used in this work.

This section was clarified to say: “Continuous data were analyzed using a Student’s t-test for parametric data or Mann-Whitney U test for non-parametric data.

  1. The discussion section is just a mere summary of the results previously described. Instead, authors should put these results in context, compare their findings with the previous publications, discussing their significance.

We have added comparisons throughout and a discussion of significance of results.

  1. Actually, this work involved a comparatively small number of pediatric patients (N = 101 in total). Therefore, these findings should be considered with extreme caution.

We have listed this as a limitation.

Reviewer 4 Report

Dear Authors,

The manuscript ID: pharmacy-1530532_v1 entitled „Intravenous versus oral step-down for the treatment of Staphylococcus aureus bacteremia in a pediatric population” written by Sarah Grace Gunter, Mary Joyce B. Wingler, David A. Cretella, Jamie L. Wagner, Katie E. Barber, and Kayla R. Stover is devoted to antistaphylococcal therapy in children.

In my opinion, the purpose of this study – assess clinical characteristics and outcomes associated with intravenous versus oral stepdown treatment of pediatric Staphylococcus aureus bacteremia, is very interesting and topical. The whole manuscript (Introduction, Materials and Methods, Results, Discussion, Conclusions) is properly organized. Introduction is concise and contains general data on bacteremia caused by S. aureus. Appropriate methods were used to carry out research. Statistical analysis was also performed The results are documented, summarized in the form of tables and correctly interpreted. Based on the results, adequate discussion and conclusions were drawn that the oral stepdown therapy appears to be a reasonable alternative to prolonged intravenous treatment for pediatric Staphylococcus aureus bacteremia, It is a well written and original review. 

According to me, this manuscript is very valuable and may be accepted for the publication in “Pharmacy”.

With highest regards,

Author Response

Thank you for your diligent review.